# An evolutionary model of rhythmic accelerando in animal vocal signalling

Yannick Jadoul [1,2,3]*, Taylor A. Hersh[2,4], Elias Fernández Domingos[3,5], Marco Gamba[6], Livio Favaro [6], Andrea Ravignani [1,2,7,8]*

1 Department of Human Neurosciences, Sapienza University of Rome, Rome, Italy, 2 Comparative Bioacoustics Group, Max Planck Institute for Psycholinguistics, Nijmegen, the Netherlands, 3 Artificial Intelligence Lab, Vrije Universiteit Brussel, Brussels, Belgium, 4 Marine Mammal Institute, Oregon State University, Newport, Oregon, United States of America, 5 Machine Learning Group, Université Libre de Bruxelles, Brussels, Belgium, 6 Department of Life Sciences and Systems Biology, University of Turin, Turin, Italy, 7 Center for Music in the Brain, Department of Clinical Medicine, Aarhus University, Aarhus, Denmark, 8 Research Center of Neuroscience "CRiN-Daniel Bovet", Sapienza University of Rome, Rome, Italy

* yannick.jadoul@uniroma1.it (YJ), andrea.ravignani@uniroma1.it (AR)

## Abstract

Animal acoustic communication contains many structural features. Among these, temporal structure, or rhythmicity, is increasingly tested empirically and modelled quantitatively. Accelerando is a rhythmic structure which consists of temporal intervals increasing in rate over a sequence. Why this particular vocal behaviour is widespread in many different animal lineages, and how it evolved, is so far unknown. Here, we use evolutionary game theory and computer simulations to link two rhythmic aspects of animal communication, acceleration and overlap: We test whether rhythmic accelerando could evolve under a pressure for acoustic overlap in time. Our models show that higher acceleration values result in a higher payoff, driven by the higher relative overlap between sequences. The addition of a cost to the payoff matrix models a physiological disadvantage to high acceleration rates and introduces a divergence between an individual's incentive and the overall payoff of the population. Analysis of the invasion dynamics of acceleration strategies shows a stable, non-invadable range of strategies for moderate acceleration levels. Our computational simulations confirm these results: A simple selective pressure to maximise the expected overlap, while minimising the associated physiological cost, causes an initially isochronous population to evolve towards producing increasingly accelerating sequences until a population-wide equilibrium of rhythmic accelerando is reached. These results are robust to a broad range of parameter values. Overall, our analyses show that if overlap is beneficial, emergent evolutionary dynamics allow a population to gradually start producing accelerating sequences and reach a stable state of moderate acceleration. Finally, our modelling results closely match empirical data recorded from an avian species showing rhythmic accelerando, the African penguin. This shows the productive interplay between theoretical and empirical biology.

**Data availability statement:** The manuscript contains no data. All scripts/code appear as Supporting information.

**Funding:** YJ, TAH, and AR were supported by Independent Max Planck Research Group Leader funding to AR. YJ and AR are also supported by the HFSP research grant RGP0019/2022. TAH is also supported by a L'Oréal USA For Women in Science Fellowship. EFD is supported by an FNRS grant for chargé de recherche (nr. 40005955). The funders had no role in study design, data collection and analysis, decision to publish, or preparation of the manuscript.

**Competing interests:** The authors have declared that no competing interests exist.

## Author summary

Animal acoustic communication is often structured in time; i.e., some animal sounds are rhythmic. Among all rhythms, accelerando occurs when the time between sounds shortens as a sequence progresses. In humans, we see accelerando for example in music. In other animals, we find accelerando in multiple species, including African penguins. Why is this particular vocal behaviour present in diverse animal lineages? How did it evolve? We use quantitative tools, namely game theory and computer simulations, to link accelerando to a specific rhythmic feature of groups, namely acoustic overlap. In particular, we test whether rhythmic accelerando could evolve under a pressure for acoustic overlap in time. We show that, when acoustic overlap is evolutionarily advantageous, simulated individuals will produce vocalisations with accelerando. To achieve overlap, this strategy to accelerate can be stable for a range of parameter values. Our computational simulations show that a population of individuals vocalising isochronously, i.e., very regularly like metronomes, can evolve to produce increasingly accelerating sequences until all individuals show accelerando. In brief, species for which vocal overlap is beneficial should evolve towards producing accelerating sequences. This, we speculate, is what may have happened to an ancestor of African penguins, resulting in the accelerando shown in this species today.

## Introduction

Many animal vocal displays have a rhythmic structure. In many cases, this structure consists of rhythms with a single underlying tempo (i.e., isochrony, with events equally spaced in time), yet several species also produce sequences with an accelerating tempo. Isochrony is the simplest and best-described rhythmic pattern in animal vocal signalling. Gibbons show it while singing, and male rock hyraxes (*Procavia capensis*) have higher reproductive success the more isochronously they vocalize [1–3]. In male orangutans (*Pongo pygmaeus wurmbii*), isochrony is present in long calls, with rhythmic regularity at various levels using different unit types and subsequences [4]. Beyond isochrony, accelerando is a less common but more complex rhythmic pattern. Accelerando is found in the songs of the field sparrow (*Spizella pusilla*), ruffed grouse (*Bonasa umbellus*), and thrush nightingale (*Luscinia luscinia*), for instance [5,6]. Beyond birdsong, vocal accelerando also occurs in various mammalian species; for example, hammer-headed bats (*Hypsignathus monstrosus*), rock hyraxes (*Procavia capensis*), and yellow-bellied marmots (*Marmota flaviventris*) have all been observed to produce accelerating vocalisations [7–9]. However, how and why this particular vocal behaviour is widespread in multiple animal lineages and contexts is so far unknown. How can vocal accelerando evolve? Most quantitative and evolutionary modelling in animal rhythmic signalling has focused on only two patterns: individual isochrony and group synchrony, i.e., when multiple individuals rhythmically overlap.

Indeed, a key aspect of concurrent displays is the amount of overlap between the vocalisations of two or more animals. In these cases, sounds produced in a shared medium, in the same frequency range, and at the same time inevitably interfere and (partially) obscure each other. Here, the precise timing of vocalisations in a display can be crucial to achieve successful communication. Three broad categories exemplify how an animal can time its vocalisations: random timing, avoidance, and overlap. Random timing, in which animals do not adapt the timing of their vocalisations in any way, could be considered the base case [10]. A strategy of avoidance entails that animals try to ensure minimal interference of their vocalisation with others and is a known mechanism for advertising identity or mating quality in several species [e.g., 11–17]. Crucially, within a species, overlap and avoidance might even coexist and serve different functions [18–20].

A less obvious communication strategy occurs when animals time their vocalisations to *overlap more* than random. At first sight, it may seem counter-intuitive that having an individual's vocalisation obscured by other vocalisations would benefit that individual. While empirical work has shown that signals within individuals of a species often overlap in time [e.g., vocal lek behavior in birds; 21,22], overlap and synchrony can frequently, and rightfully, be explained away as an undesirable epiphenomenon of precedence effects [23]. Though individuals – including potential mates – are sensitive to the overlap of conspecifics [24]; why should individuals care to detect synchrony if it were only an epiphenomenon? One hypothesis even suggests that human rhythmic abilities may have been shaped by evolutionary pressures for synchronous overlap [25]. If one hypothesizes that overlap confers an advantage, rather than a disadvantage, to individuals, which individual rhythmic patterns can evolve?

Here, we probe an evolutionary link between two rhythmic aspects of animal communication by testing whether rhythmic accelerando could evolve under pressure for acoustic overlap. We create a game theoretical model of isochronous and accelerating sequences to investigate how accelerando affects the temporal interactions and overlap between vocalising individuals. In particular, we investigate how an evolutionary pressure for maximising overlap may shape individual sequences. Crucially, our model differs from most previous work in three ways, potentially filling three gaps in the literature. First, while most models try to show the emergence of synchrony based on assumptions about individuals, in our model synchrony is the main assumption. Second, we cautiously explore the hypothesis that overlap may be beneficial, rather than detrimental. Third, to our knowledge for the first time, we model rhythmic accelerando rather than isochrony.

Linking our model to biological reality, we draw inspiration from the vocal display of a species showing both acceleration and overlap: the songs of African penguins. African penguins (*Spheniscus demersus*) are a species of banded penguin living in dense colonies along southern African coasts and islands. During breeding periods, male penguins produce a vocal display called the ecstatic display song (EDS), a sequence of short (0.1–0.4 s) vocalisations which gradually increase in tempo and intensity and lead up to a long (0.5–2 s), loud vocalisation [26,27]. These vocal displays seem to play a role in nest territory defense, mate recognition, and mate attraction, but how they bolster or facilitate these behaviours is unknown [26–29]. In the current work, these EDSs provide a starting point for a model of accelerando, as they are indeed rhythmically accelerating [30]. Moreover, African penguin EDSs are rarely produced in isolation; instead, multiple animals sing at the same time, leading to a dynamic, noisy soundscape with ample opportunity for overlap between penguins' vocalisations [31,32].

There is an additional reason why African penguin songs are an interesting case study for such an evolutionary model: Across all penguin species, only the four species of banded penguins are known to produce accelerating vocal displays; species in other genera only produce isochronous vocal displays [33]. Banded penguin species are philopatric and show high nest fidelity over consecutive seasons. Arguably, they are the most territorial among all penguin species. As banded penguins have evolutionary pressures related to territoriality and accelerating displays are absent in other penguin species, this suggests that, within the banded penguin lineage, accelerando likely evolved from the isochronous displays seen in most other penguin species. These three combined aspects – accelerating vocal displays, dense colonies with overlapping vocalisations, and a probable evolutionary trajectory from isochrony – make African penguins' vocal displays a source of inspiration for an evolutionary model of accelerando and overlap.

And so, while we do not intend to explain the evolution of African penguins' vocal behaviour in particular, the known rhythmic characteristics of these specific songs provide inspiration for a general model of accelerating sequences. Likewise, our model is agnostic with respect to the specific function or behavioural task. While in nature, the behavioural context for such vocal displays might be as diverse as electing and localising a potential mate from afar, communicating in a social group, or agonistically competing with other individuals [12,16,34–36], the model described below can apply any such context where overlap could be hypothetically beneficial to an individual.

## Methods

### Model of accelerating sequences and overlap

We modelled an accelerating vocal display as a sequence of $n$ syllable onsets (see Fig 1 insets), separated by $n-1$ inter-onset intervals (IOIs). The acceleration between two consecutive IOIs, $IOI_i$ and $IOI_{i+1}$, is calculated as $a_i = \frac{IOI_i}{IOI_i + IOI_{i+1}}$ [following 30,37–40]. Note that an acceleration rate $a$ of 0.5 corresponds to isochrony, while a higher value indicates an accelerating tempo. In our model, all sequences had the same initial tempo (i.e., $IOI_1 = 1$) and the acceleration was kept constant throughout sequences (i.e., all values $a_i = a$), such that the duration of each IOI is reduced by a constant fraction. At

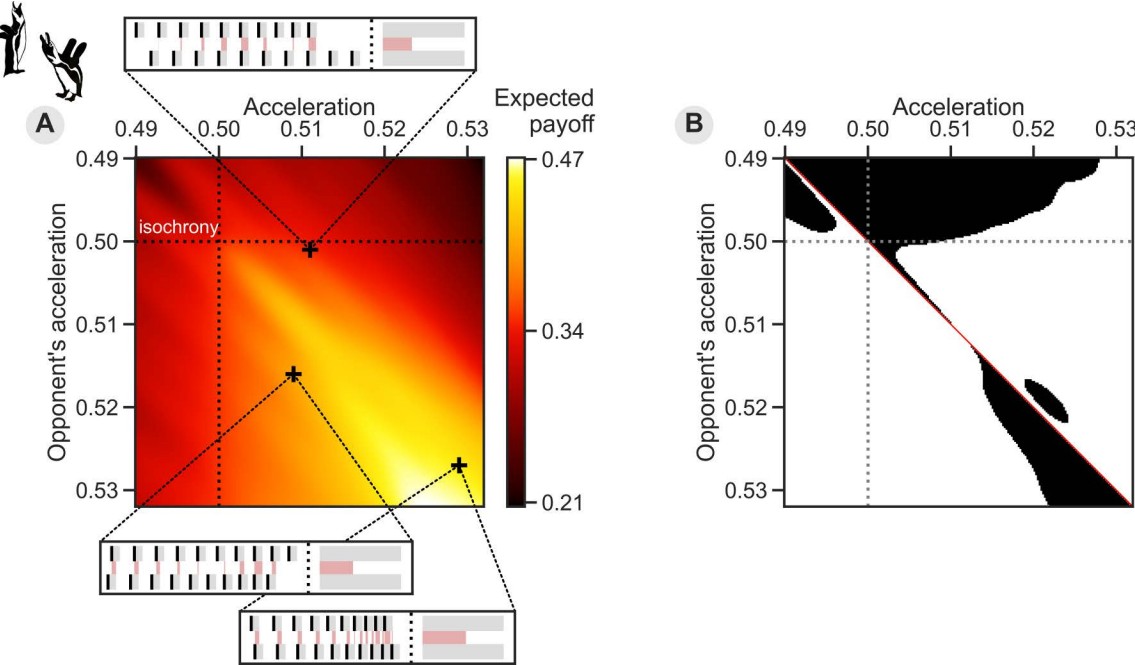

**Fig 1. The game theoretical model of the interaction between two individuals with different levels of vocal accelerando. (A)** The continuous payoff matrix for each pair of possible acceleration values in the evolutionary game theory model shows a higher expected payoff for higher acceleration rates (colour indicates amount of payoff). Note however that because of the cost associated with higher acceleration, the payoff matrix is asymmetrical (i.e., the region with the highest payoff is achieved by an acceleration rate of 0.52 if the other individual accelerates at rate 0.53). The three insets show example pairs of modelled accelerating sequences and the calculation of relative amount of overlap: Black bars are syllable onsets, followed by the syllable durations in grey, while red patches in between the two sequences indicate intervals where the two sequences overlap. The bars on the right show the total time spent vocalising and the relative amount of overlap between the two sequences. Crosses indicate the corresponding point in the payoff matrix for two given acceleration rates in an inset. An interactive demonstration of the modelled accelerating sequence and their overlap is available in S1 Code. **(B)** The pairwise invasibility plot resulting from this payoff matrix shows which acceleration strategies (in black) have a higher payoff than the diagonal's payoff on the same horizontal line. An individual with such an off-diagonal strategy would receive a higher payoff than one with the same strategy as its opponent. Consequently, this strategy has an evolutionary benefit and can invade a homogenous population. Conversely, as there are no black regions to the left and the right of the diagonal for an opponent with an acceleration strategy around 0.51, such a moderate acceleration rate is evolutionary stable.

each resulting onset, we model a syllable with constant duration $d$. These modelling choices were based on previous empirical analyses of African penguin EDSs, which found acceleration and syllable duration to not strongly vary within displays [30]. Altogether, our model captures the main characteristics of an accelerating sequence in three simple parameters.

In subsequent analyses, we varied sequences' acceleration $a$ to study the effect of different accelerations on the overlap between sequences (see below), while syllable count $n = 10$ and syllable duration $d = 0.36$ were chosen to approximate the values in the empirical African penguin data [27,30]. Given these values for the maximal acceleration $a_{max} \approx 0.532$ (see Method A in S1 Appendix).

Next, given two accelerating sequences with acceleration rates we calculated the expected amount of overlap $O_{a,b}$ between two sequences. Assuming that two individuals do not have precise control over the relative phase of their sequences, the overlap between two sequences is averaged over all relative phase offsets between the two sequences (see Method B and Fig A in S1 Appendix). Additionally, we associated a cost $c_a$ with each acceleration rate $a$, assuming that a higher acceleration is more intense to produce (e.g., a higher physiological cost; see Method B in S1 Appendix) and that any perceptual cost for a listener is negligible in comparison.

In conclusion, when aiming to maximise the overlap while minimising the physiological cost, the total expected payoff of a modelled sequence with acceleration $a$ with respect to another acceleration rate $b$ is $\Pi_{ab} = O_{a,b} - c_a$.

## Evolutionary game-theoretical model

Based on these abstract representations of accelerating sequences and their mutual overlap and payoff, we created a simple evolutionary game-theoretical model to investigate the evolutionary advantage of accelerating strategies. In this first model and analysis, we assume a discrete set of possible accelerations and aim to find which are evolutionarily stable. Later, we relax this constraint and investigate the evolutionary dynamics of a continuous set of possible accelerations (see next subsection).

We describe the dynamics between different acceleration rates through an evolutionary process [41] where a finite population of individuals is under a selective pressure proportional to the fitness of each acceleration strategy (i.e., an individual's fixed rate of acceleration). The population evolves as individuals get replaced one-by-one by new individuals with a higher fitness strategy (for details, see Method C in S1 Appendix). Thus, our model considers the strategic interplay of acceleration strategies and selective pressure towards increasing overlap to explain how accelerando could develop in a vocal display. Specifically, the selective pressure is determined by considering the payoff between each pair of acceleration rates $\Pi_{ab}$.

We implemented this analytical model using the EGTtools Python package [version 0.1.10; 42]. We analysed the invasion dynamics (under the small mutation limit; see Method C in S1 Appendix) of a subset of 21 discrete acceleration strategies, with accelerations equally spaced from 0.49 to 0.53. This analysis was done for population size $Z = 100$ and selection intensity $\beta = 1$.

## Numerical simulations

Following the above game-theoretical model, we simulated the evolutionary trajectory of a finite population of individuals, each with its own fixed acceleration. Additionally, in this computational simulation, we removed the restriction of the finite set of acceleration strategies and allowed any acceleration strategy in the range from 0.490 to 0.532 (i.e., from a slight deceleration to the maximum acceleration $a_{max}$). Within such a population, we calculate each individual's fitness as its expected payoff, interacting uniformly with all others. Starting with a population of $Z$ isochronous individuals, each of the $Z$ individuals in the subsequent generation will inherit an acceleration rate based on the fitness of the previous generation. In addition, a random mutation will be added to each new individual, slightly increasing or decreasing each inherited acceleration rate (see Method D in S1 Appendix).

Below, we focus on the results of 100 simulations of $10^6$ steps, each with a population size of $Z = 100$, selection intensity $\beta = 1$, and mutation size $\sigma = 10^{-4}$. However, we also ran simulations for a range of parameter combinations of $Z$, $\beta$, and $\sigma$ to test the sensitivity of the results to parameter variation.

## Results

The initial analysis of the calculated payoff matrix shows that higher acceleration values in our model result in a higher payoff, driven by the higher relative overlap between sequences (Fig 1). Importantly, for high acceleration values, the modelled cost shifts the maximum payoff off-diagonal, incentivising individuals to accelerate less than others in the population. This cost gives rise to more complex evolutionary dynamics since it introduces a divergence between an individual's incentive and the overall payoff of the population.

Theoretical analysis of the invasion dynamics (under the small mutation limit) of the discrete 21 acceleration strategies shows a convergent stable, non-invadable range of strategies [43] for moderate acceleration levels (0.506 to 0.514; Fig 2). These strategies account for about 58% of the stationary probability distribution. Only about 5% of the stationary distribution has a lower level of acceleration than 0.506 (i.e., close to isochrony).

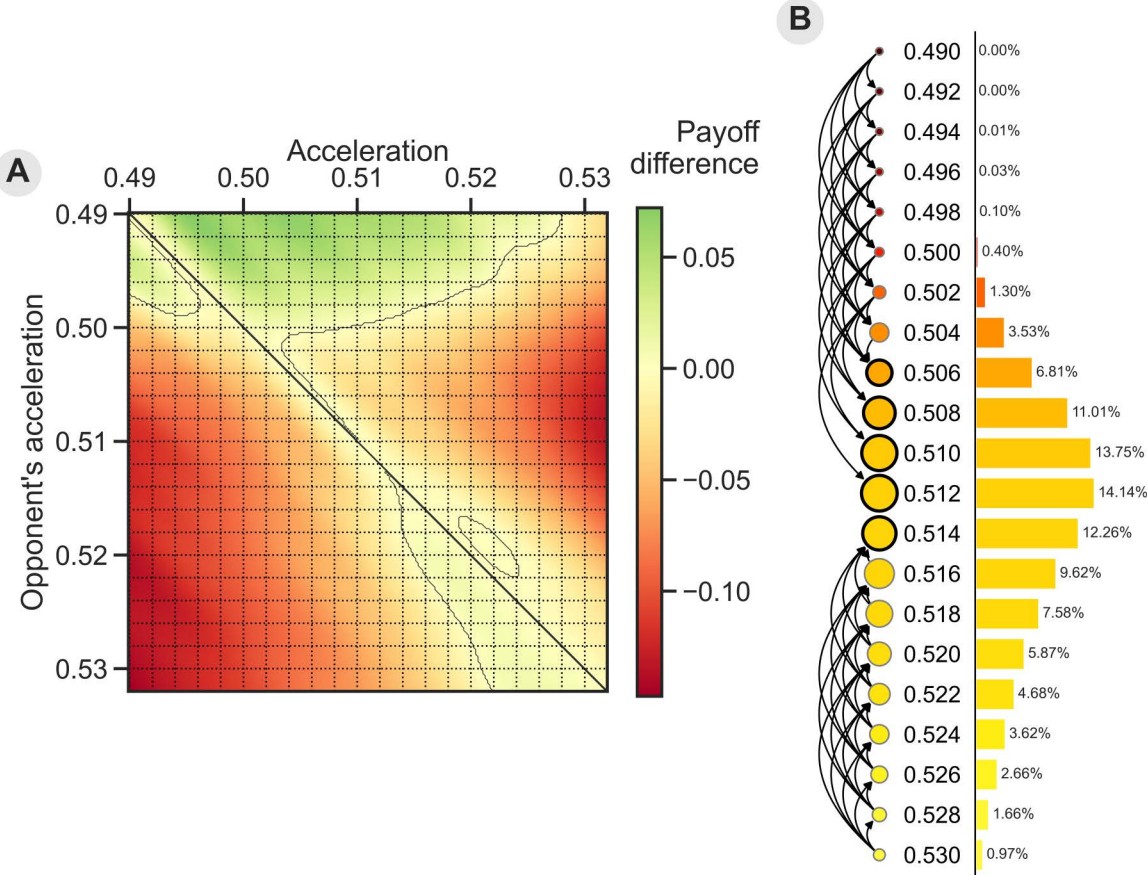

**Fig 2. Theoretical analysis results of the model's invasion dynamics and evolutionary stable strategies. (A)** Extending the pairwise invasibility plot (Fig 1) with the difference in payoff indicates the intensity of selective pressure in a homogeneous population. As before, areas where a strategy's payoff is higher than the corresponding payoff on the diagonal indicate that a homogeneous population can be invaded by a mutant strategy. The horizontal and vertical dashed lines show the 21 discrete acceleration strategies whose invasion dynamics were analysed. **(B)** Under the small mutation assumption, the invasion diagram between strategies for 21 discrete acceleration values shows a range of evolutionary stable strategies from 0.506 to 0.514, with a probability of 58% in the stationary distribution. Acceleration rates close to isochrony have very low probability in the stationary distribution: Acceleration rates of less than 0.506 account for only 5% of the total probability, indicating how unlikely they are to evolve and persist in a population. The invasion diagram shows which strategies can be invaded by which others (arrows are only drawn between acceleration rates that differ by 0.1 or less). To the right of this graph, the stationary distribution (under the small-mutation hypothesis) quantifies likely and unlikely acceleration strategies for a population to evolve.

Our computational simulations confirm these results: A simple selective pressure for maximising the expected overlap (while minimising the associated physiological cost) causes an initially isochronous population to evolve towards producing increasingly accelerating sequences until the population reaches equilibrium around a moderate rate of acceleration. Fig 3 shows the results of 100 different runs of $10^6$ steps and a final distribution that matches the results of our previous analysis of the invasion dynamics (see Fig 2). Moreover, simulations with different values for $Z$, $\beta$, and $\sigma$ produce analogous results (Fig 4), demonstrating the exact parameter values of the evolutionary simulation do not change these conclusions.

Overall, these analyses show that, under our straightforward modelling assumptions, accelerating sequences have a higher level of expected mutual overlap. If overlap is beneficial for an individual, emergent evolutionary dynamics allow an isochronous population to gradually start producing accelerating sequences and reach a stable state of moderate acceleration. Intermediate, rather than maximum, levels of acceleration constitute a stable equilibrium which maximises the achieved overlap in a population given an individual's cost.

## Discussion

Our theoretical models and evolutionary simulations describe a hypothetical evolutionary scenario under which accelerando can develop in a vocal display: If overlap benefits individuals, accelerating sequences are a controlled way of maximising vocal overlap among competing signallers. In addition, the outcome of our model dovetails with empirical African penguin data: The range of evolutionary stable acceleration rates resulting from our model broadly matches the acceleration observed in this species' displays [30]. While the current model does not rule out other possible evolutionary explanations for the penguins' accelerando, it quantifies a path from isochrony to accelerando. Furthermore, the agreement between the real-world data and our model results highlights the potential of African penguins as a model species for investigating accelerando.

Like any model, these conclusions are subject to several assumptions. Our first assumption is that overlap can be evolutionarily advantageous in a particular context or ecological setting. In numerous animal species, vocal signals from conspecifics do overlap at millisecond-to-second timescales [24]. Individuals are often sensitive to overlaps and, when

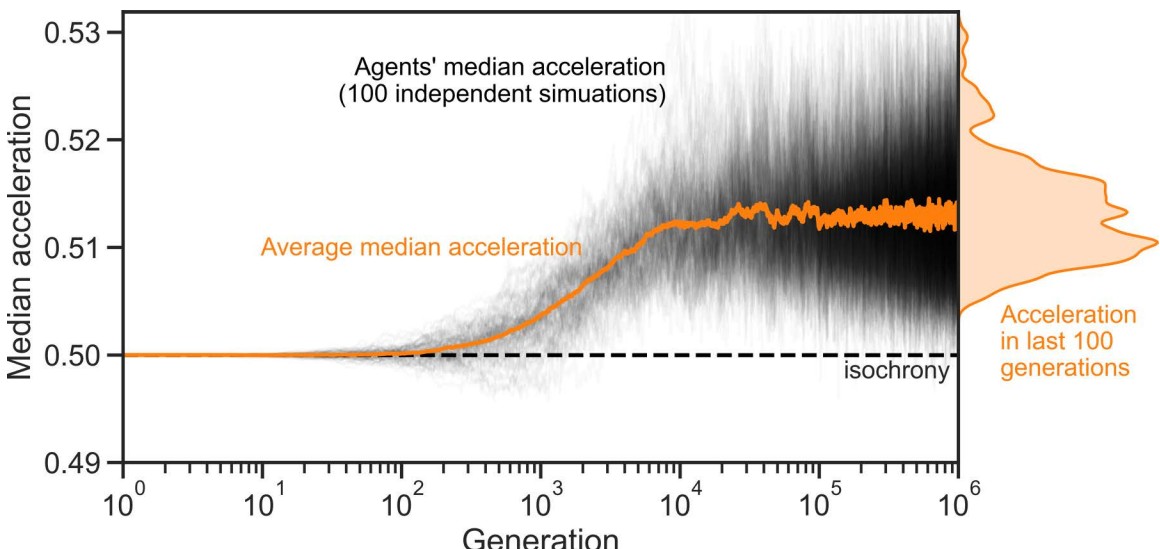

**Fig 3. Computational simulation results of the model's evolutionary dynamics.** In our numerical simulations, the median acceleration in a population of initially isochronous individuals quickly rises to around 0.513. The distribution of the individuals' acceleration in the last 100 generations of all 100 simulation runs approximates the stationary distribution in Fig 2.

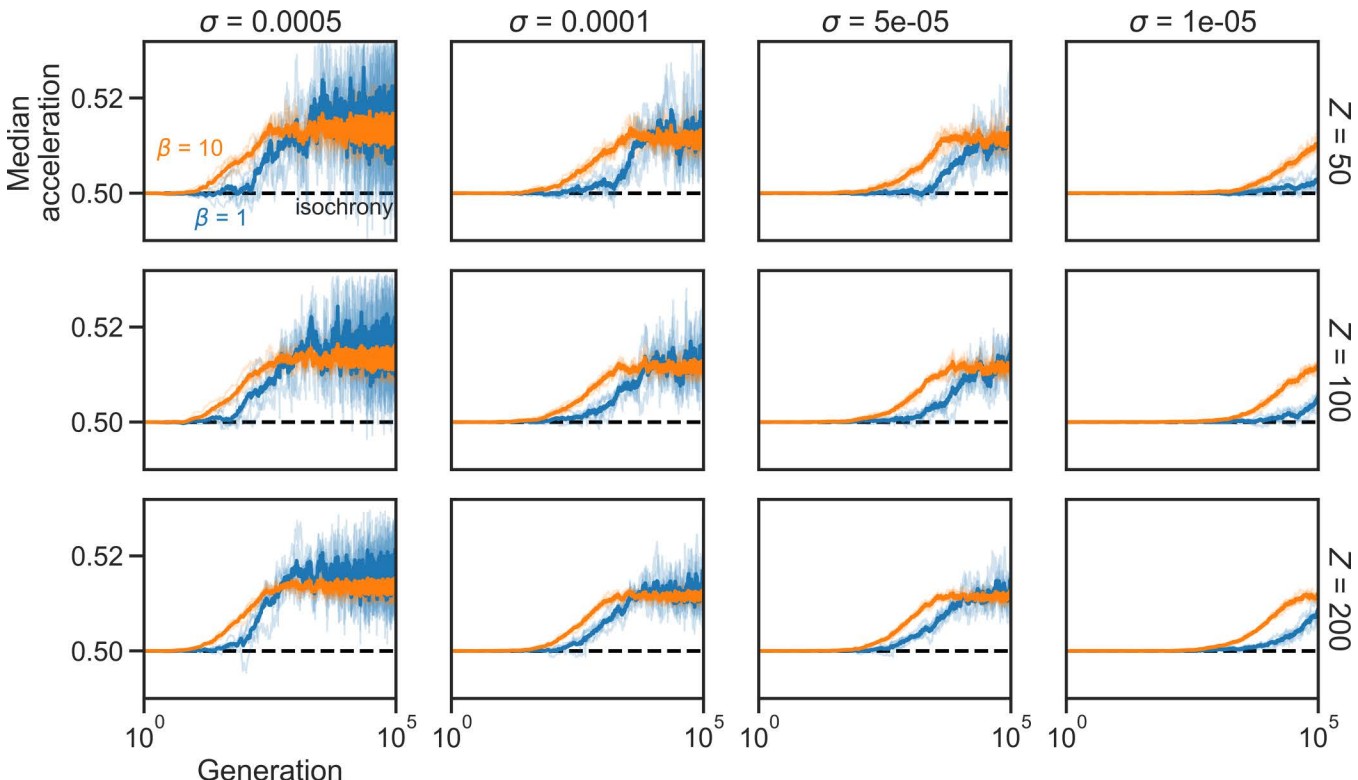

**Fig 4. Sensitivity analysis of evolutionary simulation parameters.** Evolutionary simulation ($10^5$ steps) with different combinations of parameter values for population size $Z \in \{50,\ 100,\ 200\}$ (rows), selection intensity $\beta \in \{1,\ 10\}$ (line colours), and mutation size $\sigma \in \{5e^{-4}, 1e^{-4},\ 5e^{-5},\ 1e^{-5}\}$ (columns) demonstrate that while these parameters affect the evolutionary trajectory, the simulated populations still evolve to produce moderately accelerating sequences (i.e., the results in Fig 3 are robust). For each parameter combination, the population's median acceleration of five independent simulations is shown, as well as their average.

choosing mates, can discriminate between leader and follower [44]. Unlike our paper, most modelling work in animal rhythm to date has focused on understanding how overlap *avoidance* leads to acoustic alternation or synchrony [12–14]. Such work often asks: Given individual rhythmic strategies, which group rhythms can emerge? Here, we ask the opposite question: Which individual rhythmic strategies emerge when there is pressure to overlap? Under such pressure – an assumption surely disputed by some – we show that, in fact, a rhythmic pattern seen in actual avian species, accelerando, does emerge.

Our second assumption is that a higher rate of acceleration and a faster tempo are more costly to an individual. This assumption is physiologically plausible and supported by cross-species literature. In multiple taxonomic groups, syllable rate and proportion of time spent phonating, rather than total vocal display length, positively correlate with metabolic costs [45–47]. We only included one parametrisation of physiological costs in our analyses; future work should generalise and compare multiple cost functions so that one can choose the most plausible cost estimate for the species of interest.

Additional implicit assumptions concern the vocal capabilities of the individuals. In particular, our simulated individuals have no control over their relative phase; i.e., they cannot adjust a vocalisation's delay with respect to each other. In addition, our simulated individuals cannot change the (initial) tempo of their vocalisation; i.e., they cannot evolve to produce faster isochronous sequences instead of gradually accelerating ones. Crucially, our simulated individuals are only allowed to decelerate or accelerate, and no other "rhythmic choices" are possible. In hindsight, this can intuitively explain the increased amount of overlap in accelerating sequences: Accelerating rhythms will have a higher duty cycle

and, therefore, a higher chance of overlap. Our model explicitly quantifies the interaction of this effect and the assumption of random offsets between sequences. One way of further understanding this is by realising isochrony is a risky strategy if there is lack of control over the phase offset. While two isochronous sequences can end up with close to 100% overlap, an unlucky offset will result in zero overlap. Based on our results, accelerando could be viewed as a strategy to compensate for the lack of control over phases. The results also highlight the potential importance of control over the phase offset between two vocalisations. Future work should model different levels of control and study how they influence the resulting evolutionary dynamics.

Another crucial piece of context to interpret our results is that the evolving traits of individuals in a population directly impact the fitness landscape. While particular evolutionary adaptations will have a clear benefit, other evolutionary advantages are dependent on the rest of the population. The model here falls in the latter category, as the fitness of a certain acceleration strategy directly depends on the proportion of other strategies present in the population. In Fig 1, the maximum payoff for an individual is achieved with an acceleration of around 0.525, but *only if* the rest of the population accelerates at rates around 0.53. This is not the acceleration rate we see at equilibrium (Figs 2 and 3). At high acceleration rates, the increased fitness of a *slightly* lower rate would over time result in more individuals accelerating at this lower rate. As the whole population's acceleration rate changes over time, so does the evolutionary landscape and maximally achievable payoff. As such, while our model might seem simple and its results perhaps obvious, they comprise more than the mere tradeoff between overlap and cost faced by each individual. Evolutionary models and computational simulations are indispensable to capture the complex feedback loop between individual strategies and overall population effects, even when the results appear intuitive.

Taken together, we believe our model's proof of concept remains valid: acceleration can emerge from overlap. Future work can remove or relax the necessary assumptions by gradually building on the current model. In particular, future models should explore different ways temporal structure in vocalisations could provide an evolutionary benefit, e.g., by implementing different types of sequences or different payoff matrices, and comparing how they affect the evolutionary dynamics and simulations. Our model could also be extended to allow for more temporal characteristics, other than just a single acceleration parameter, to vary between individuals. For example, what happens if the number of syllables in each sequence can change or when the initial IOI length can vary? Or what if individuals have a certain amount of vocal flexibility during an interaction, and can adapt to each other's rhythmic patterns? Alternative hypotheses can be implemented in the same evolutionary game theory framework and compared to empirical data on accelerando. Besides accelerando, other individual rhythmic patterns could emerge (e.g., nested isochrony, or rhythms with intervals following small integer ratios); future work should select and compare key ones. Vice versa, the assumptions and restrictions of the current model and the results they produce show that a more complex mechanism for synchronisation is not strictly necessary to evolve the production of accelerating vocal displays. There might be many different evolutionary paths to accelerando and other rhythmic structures.

As such, our results are not a conclusive or exhaustive explanation of the evolution of accelerando in vocal displays. Rather, our current model is a first proof of concept laying the groundwork for future exploration of the phenomenon in three ways: It provides a simple framework to investigate accelerando and rhythm in animal vocalisations, it delivers the first evolutionary explanation of how rhythmic accelerando could emerge, and it generates concrete hypotheses that can be empirically tested in living animals. In particular, our model predicts species with accelerating vocal displays, such as banded penguins, will have a higher-than-average overlap between these displays. Observational data should be collected to test this hypothesis, and behavioural experiments should target the relationship between overlap and an individual's fitness. Such data will, in turn, inform and drive the development of more realistic models.

Modelling approaches can complement empirical work by simulating how rhythmic displays from multiple individuals interact under different evolutionary pressures. Our approach can also be used to analyse accelerating acoustic displays that are produced via non-vocal means, such as substrate drumming by some birds; e.g., woodpeckers [48], ruffed grouse

[49], primates, [50]. By merging cross-species theoretical and empirical research, we can continue progressing our understanding of animal vocal rhythms' evolution.

## Supporting information

**S1 Appendix. An evolutionary model of rhythmic accelerando in animal vocal signalling. Fig A.** Relative approximation error estimates of payoff values. **Method A.** Maximum acceleration rate **Method B.** Calculation of expected relative overlap and expected payoff. **Method C.** Evolutionary process and stationary distribution. **Method D.** Evolutionary selection and mutation in numerical simulations.
(PDF)

**S1 Code. Models and simulations code.**
(ZIP)

## Acknowledgments

We thank Giardino Zoologico di Pistoia, Zoomarine Roma, and Zoom Torino for facilitating research and providing access to African penguin colonies; Stephanie King, Dan Stowell, and Christian Herbst for feedback that improved the manuscript.

## Author contributions

**Conceptualization:** Yannick Jadoul, Livio Favaro, Andrea Ravignani.

**Formal analysis:** Yannick Jadoul, Elias Fernández Domingos.

**Investigation:** Yannick Jadoul.

**Methodology:** Yannick Jadoul, Elias Fernández Domingos, Andrea Ravignani.

**Software:** Yannick Jadoul.

**Supervision:** Livio Favaro, Andrea Ravignani.

**Validation:** Marco Gamba.

**Visualization:** Yannick Jadoul, Elias Fernández Domingos.

**Writing – original draft:** Yannick Jadoul, Taylor A. Hersh, Elias Fernández Domingos, Andrea Ravignani.

**Writing – review & editing:** Yannick Jadoul, Taylor A. Hersh, Elias Fernández Domingos, Marco Gamba, Livio Favaro, Andrea Ravignani.

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
