## [Decision Letter · Decision Letter 0]

19 Dec 2024

PCOMPBIOL-D-24-01329

An evolutionary model of rhythmic accelerando in animal vocal signalling

PLOS Computational Biology

Dear Dr. Ravignani,

Thank you for submitting your manuscript to PLOS Computational Biology. After careful consideration, we feel that it has merit but does not fully meet PLOS Computational Biology's publication criteria as it currently stands. Therefore, we invite you to submit a revised version of the manuscript that addresses the points raised during the review process.

Please submit your revised manuscript within 60 days Feb 18 2025 11:59PM. If you will need more time than this to complete your revisions, please reply to this message or contact the journal office at ploscompbiol@plos.org. Please include the following items when submitting your revised manuscript:

We look forward to receiving your revised manuscript.

Kind regards,

Xingru Chen, Ph.D.

Guest Editor

PLOS Computational Biology

Zhaolei Zhang

Section Editor

PLOS Computational Biology

**Journal Requirements:**

At this stage, the following Authors/Authors require contributions: Yannick Jadoul, Taylor Hersh, Elias Fernández Domingos, Marco Gamba, Livio Favaro, and Andrea Ravignani. Please ensure that the full contributions of each author are acknowledged in the "Add/Edit/Remove Authors" section of our submission form.

5) We have noticed that you have uploaded Supporting Information files, but you have not included a list of legends. Please add a full list of legends for your Supporting Information files after the references list.

6) We are unable to open the following Supporting Information file: code.zip. Please kindly revise as necessary and re-upload.

7) Some material included in your submission may be copyrighted. According to PLOSu2019s copyright policy, authors who use figures or other material (e.g., graphics, clipart, maps) from another author or copyright holder must demonstrate or obtain permission to publish this material under the Creative Commons Attribution 4.0 International (CC BY 4.0) License used by PLOS journals. Please closely review the details of PLOSu2019s copyright requirements here: PLOS Licenses and Copyright. If you need to request permissions from a copyright holder, you may use PLOS's Copyright Content Permission form.

Potential Copyright Issues:

- Figure 1. Please confirm whether you drew the images / clip-art within the figure panels by hand. If you did not draw the images, please provide a link to the source of the images or icons and their license / terms of use; or written permission from the copyright holder to publish the images or icons under our CC BY 4.0 license. Alternatively, you may replace the images with open source alternatives. See these open source resources you may use to replace images / clip-art:

8) We note that your Data Availability Statement is currently as follows: "The manuscript contains no data. All scripts/code appear as supplement.". Please confirm at this time whether or not your submission contains all raw data required to replicate the results of your study. Authors must share the “minimal data set” for their submission. PLOS defines the minimal data set to consist of the data required to replicate all study findings reported in the article, as well as related metadata and methods (https://journals.plos.org/plosone/s/data-availability#loc-minimal-data-set-definition).

- The points extracted from images for analysis..

**Reviewers' comments:**

Reviewer's Responses to Questions

**Comments to the Authors:**

Reviewer #1: This manuscript is about an EGT model of the dynamics of accelerando as an evolving trait in a population of vocalizing animals. In the model, fitness (i.e., payoff) is operationalized by the net overlap of the sequences of two interacting individuals. There is a trade-off between acceleration and production effort, which eventually leads to the existence of an evolutionarily stable strategy representing a stable amount of accelerando in the population (actually, in ‘evolutionary invasion analysis’ terminology it’s a convergent stable strategy, McGill et al. 2007).

I know relatively little about animal communication, so I cannot say much about the relevance and soundness of this study in this regard. For what it's worth, the African penguin example looks good to me. However, as far as the presented modeling work is concerned, I found this to be an excellent manuscript. The model is reasonably elegant, clearly described, as are its results, and the discussion is very insightful (just a few remarks, see below). The supplementary document is very informative (in fact, some of the information could be moved to the main manuscript). Overall, I have little to complain about, but I do have a couple of suggestions and/or questions that the authors might want to consider.

* Payoff is defined as overlap minus production costs. Crucially, production only affects one individual, which in turn leads to asymmetries in the payoff matrix, as the authors point out. I guess, perceptual costs exist as well (because sequences with very short pauses between syllables might be difficult to decompose). Is there a reason why perceptual costs are not modeled? Or is the argument that perceptual costs are very small compared to articulatory costs?

* Maybe say in the paragraph starting with “Next, given two accelerating sequences…” that phases between the two sequences are determined randomly. It is specified in S2, I know, but this information would have helped at this point in the main manuscript.

* Figure 1 is good, but for the purpose of this manuscript, the pairwise invasibility plot (PIP) computed in the notebook “Models and simulations” (i.e., the black & white plot created in “In [9]”) is much more informative than the continuous payoff matrix. This is because from the PIP one can immediately infer that there is an attracting ESS (i.e., a continuously stable strategy) in the middle of the trait space. And one can see that the isochrony strategy does not even represent an evolutionary equilibrium. I strongly recommend putting this plot next to the red-yellow one in Fig 1 since there is some space left anyway.

* Another thought on the PIP if you think of it in terms of evolutionary invasion analysis (i.e., assuming that mutations are so rare that new mutants always face a homogeneous resident population – Brännström et al. 2013): Interestingly, it displays two islands (one in the lower left corner, above the diagonal, and one in the upper right quadrant below the diagonal). I understand that the authors are interested in small mutational steps of accelerando, but as a matter of fact, this payoff matrix allows for rock-paper-scissor dynamics due to these islands if larger jumps are allowed. That is, theoretically, the evolutionary dynamics could get stuck in a loop of alternating high-level accelerando strategies without approaching the ESS in the middle. See Gyllenberg and Service (2011) for more details. I find this is actually an interesting scenario, but I leave it for the authors to decide in how far they want to comment on this.

* Discussion section, “If overlap benefits individuals, accelerating sequences are a controlled way of maximising vocal overlap among competing signalers.”: I think a crucial point also is that this holds true, on average, *even if there are phase differences* (i.e., if both do not start signaling at the same time). I can imagine that isochrony is a risky strategy if there is lack of control about phases because if you’re not lucky you end up having zero overlap (on the other hand, if you *are* lucky, you might have 100% overlap, so on average it might balance out). This brings me to the following question: how central is the assumption of having no control over the relative phase for the model/analysis? Could accelerando be viewed as a strategy to compensate for the lack of control over phases?

* Paragraph starting with “Crucially, our simulated individuals are only allowed” in the discussion: I am not sure if I can buy the argument in this paragraph. At the surface, it looks like a ‘reductio ad absurdum’ line of argumentation, but it doesn’t seem to be entirely logical to me. E.g., “One may argue that a faster, purely isochronous signal could perform equally well. But if [a higher?] duty cycle could explain everything, we would find equilibria for highest levels of acceleration, rather than the intermediate acceleration we do find.” Maybe I have misunderstood it, but isn’t the whole point that the mid-level ESS emerges because of the trade-off with production costs? Obviously, fast isochronous signals should also have higher costs than not so fast ones. Similarly, “this hypothesis is difficult to test because it would practically entail comparing an infinite number of individual rhythmic patterns.” – not a convincing point in the present context since you *are* comparing an infinite number of different strategies (the accelerando trait is continuous). Finally, “Following this line of thought, there may not even be a need for rhythm, as an uninterrupted long sound could grant maximum overlap.” I mean, yes, if overlap was the only factor. But an uninterrupted long sound violates so many other fundamental assumptions (e.g., that we are talking about syllable sequences here) that I am not sure if this is a valid (valid in the sense of ‘legit absurd’) conclusion. Overall, the paragraph left me a bit puzzled.

* Article missing in “We include such [a] selective pressure …” on page 5 of the manuscript.

* Supplement, S3: How is fitness f_a defined in S3? (It is only defined for S4 below.)

References:

Brännström, Å., Johansson, J., & Von Festenberg, N. (2013). The hitchhiker’s guide to adaptive dynamics. Games, 4(3), 304-328.

Gyllenberg, M., & Service, R. (2011). Necessary and sufficient conditions for the existence of an optimisation principle in evolution. Journal of mathematical biology, 62, 359-369.

McGill, B. J., & Brown, J. S. (2007). Evolutionary game theory and adaptive dynamics of continuous traits. Annu. Rev. Ecol. Evol. Syst., 38(1), 403-435.

Reviewer #2: The MS is on the evolution of overlapping calls. It is an interesting topic; in many well-documented studies, signalling animals actively avoid call overlap, so why have some species (like the African penguin) produce calls with an accelerated call rate with and increase of overlapping calls?

The authors have made an evolutionary model of call evolution where the fitness is calculated from call overlap minus a cost (the physiological cost of call production, increasing for accelerated call rates). This model produces an increase in accelerated callrates and call overlap in the population.

As far as I see, this model is completely circular. The selection pressure benefits overlap, and accelerated calls will increase overlap(unless overlap is actively avoided), so evidently, if natural selection works as an algorithm, ( which it does), it will produce the resultfollowing from the initial hypothesis.

Another problem, I think, is that the behavioral context of the call is not addressed in the MS. In the video in the supplementary materials, we see a lot of vocal interactions of penguins with different call types. The amount of call overlap appears to be rather small in this video, at least for the ecstatic call, normally uttered by single birds (but inducing others to follow in chorus, Favaro et al. 2014) . Also, the ecstatic call is long, and overlap with other calls may simply not be very important for example for localization of the caller. Also, whereas it is clear that animals often avoid call overlap, overlap can be an indication of increased aggression in agonistic encounters. I think these are competing hypotheses - either that the overlap is incidental, and selection pressure neutral, but there could be a selection pressure for increasing duty cycle of the call, or that overlap has a direct agonistic function. In species that actively avoid call overlap, the behavioral context - of mate selection and localization - is different.

specific comments:

Introduction: I think you need to discuss the different behavioral contexts here. The behavioral task of selecting and localizing a potential mate from afar is very different from communicating in a social group and again different from the communication between two agonists. Humans without hearing impairment have no great problems with call overlap (in cocktail party-situations, for example).

l.14 from end of introduction: 'leading to a dynamic, cacophonous soundscape' - it is not necessarily cacophonous for the penguins that will have cues based on location and distance to the other penguins in the colony.

Discussion: ,l. 3 'If overlap benefits individuals...'- I think it is clear that this assumption will produce acceleration of call rate, without having to run an evolutionary model. You need to consider other hypotheses (as outlined above)

Reviewer #3: This paper sets out to demonstrate that a simple selective pressure to maximize expected overlap in vocalizations of a fixed duration, while minimizing the physiological cost, causes an initially isochronous population — that does not have any mechanism of synchrony — to evolve towards producing increasingly accelerating sequences until a population-wide equilibrium of rhythmic accelerando is reached.

I found the paper a difficult read initially, but I think I understood by the end. I simply recommend that the authors bring some of the information from the discussion into the introduction to make understanding of the model easier and more intuitive.

Caveats: I am a modeler, but I don’t work in the area of evolution. I am not very familiar with the animal literature. And I I did not have time to work through the entire model. I think I followed the basic idea, but some pretty simple questions arose for me. Perhaps addressing these questions up front will help the authors communicate better these findings with a general readership.

I’m not sure I understand the first model. If each individual has it’s own acceleration rate, then how can it be said that acceleration is evolving? It there to begin with. Is it because some rates are = .5? The mechanism for acceleration is assumed to exist and appear at random, and then accelerating individuals are selected for. It that right? This could be made clearer.

Why doesn’t synchrony (& isochrony) always result in the highest overlap? Can you answer this with an intuitive explanation. If two animals are changing tempo (in any direction) at different rates, overlap should be less than it is for perfect synchrony, right?

If animals don’t reposed to (control) relative phase, then how to they maintain any type of synchrony? They just begin together and proceed at what happens to be an identical rate?

10^6 means 100,000 generations? Is that biologically realistic?

Could this phenomenon occur if individuals were trying to synchronize, but each had a different rate, and they tried to adapt to one another’s rate?

As I read this paragraph, it addresses many of my questions in one way or another: “they cannot adjust the note delay with respect to each other. In addition, our simulated individuals cannot change the (initial) tempo of their vocalisation; i.e., they cannot evolve to produce faster isochronous sequences instead of gradually accelerating ones. The choice of these two assumptions had one main underlying reason. Because isochronous tempo and relative phase are the most common parameters in models of animal rhythms, we decided to avoid manipulating those two parameters to test whether rhythmicity could nonetheless evolve.”

Could some of these things be said sooner than in the discussion. For example, I don’t think you said, up to this point, that they begin at a random delay. If that’s true, should be said clearly sooner.

I think the statement on the limitations is reasonable and the authors are not claiming anything beyond what their results show.

Also, on page 2, line2: “emergency” should be “emergence”

**Have the authors made all data and (if applicable) computational code underlying the findings in their manuscript fully available?**

Reviewer #1: Yes

Reviewer #2: Yes

Reviewer #3: Yes

PLOS authors have the option to publish the peer review history of their article (what does this mean? ). If published, this will include your full peer review and any attached files.

**Do you want your identity to be public for this peer review?** For information about this choice, including consent withdrawal, please see our Privacy Policy .

Reviewer #1: **Yes: ** Andreas Baumann

Reviewer #2: No

Reviewer #3: No

**Figure resubmission:**
---

## [Decision Letter · Decision Letter 1]

28 Mar 2025

Dear Mr. Jadoul,

We are pleased to inform you that your manuscript 'An evolutionary model of rhythmic accelerando in animal vocal signalling' has been provisionally accepted for publication in PLOS Computational Biology.

Best regards,

Xingru Chen, Ph.D.

Guest Editor

PLOS Computational Biology

Zhaolei Zhang

Section Editor

PLOS Computational Biology

Reviewer's Responses to Questions

**Comments to the Authors:**

Reviewer #1: I would like to thank the authors for addressing the points I have made in my review so carefully. In particular, I found the updated figures to be more informative than before (Fig 2a is great!), and also the discussion section now shows a clear line of argumentation. The authors are probably right in their assessment of the relevance of the (potential) rock-paper-scissor dynamics that one could infer from the PIP; this would have probably bloated the manuscript.

I have no further comments on the manuscript.

Reviewer #4: In their original manuscript “An evolutionary model of rhythmic accelerando in animal vocal signalling,” the authors present an evolutionary game theoretical model examining evolution of call acceleration under the simple assumptions that overlap in calls is beneficial and faster acceleration has a metabolic cost.

I was asked to provide a review following initial reviews and revision, so have the privilege of reading and considering three prior reviewers’ comments and the authors’ reponses.

I am of two minds on this paper. First, I believe that evolutionary modeling such as the authors have carried out is potentially informative and beneficial, driving interesting and productive hypothesis generation and testing. I commend the authors for bringing a range of quantitative and computational approaches to bear on the field of animal communication and rhythm. I am not a modeler, but my basic understanding of the model suggests it is interesting and valid (with just a couple requests for clarification, below). Second, the paper does not meaningfully treat with preexisting theory and evidence on the biological function and mechanisms of call acceleration in different species. I take the authors’ point, in their response to the initial reviewers, that this is a modeling paper more than a vocal communication paper. But of course it is a biological modeling paper linked to biological reality (the African penguin). I think the authors are walking a fairly fine line here – while obviously modeling can be productive, and it is not reasonable to expect all models to comprehensively account for the complexity of biological reality, models will be productive and clarifying in large part based on their plausibility and the overlap of their assumptions with reality. Here I could be convinced that the authors’ current model is useful/productive, but I do think I’d be helped along by some clarifications and acknowledgements, as detailed below.

The authors suggest that the bulk of evolutionary modeling has assumed that NON overlap of calls is beneficial. The idea here is that calls carry information, and that information can be obscured when calls overlap. Obviously, that is true in many cases. But there are many perfectly accepted cases where call overlap may be beneficial, and the authors should note these with more clarity in the introduction. Perhaps most importantly, the extended literature on vocal lek behavior in birds (e.g., just the tip of the iceberg, Gibson, R. M. (1989). Field playback of male display attracts females in lek breeding sage grouse. Behavioral Ecology and Sociobiology, 24, 439-443.

Stiles, F. G., & Wolf, L. L. (1979). Ecology and evolution of lek mating behavior in the long-tailed hermit hummingbird. Ornithological monographs, (27), iii-78.).

Here, overlapping calls have been suggested to A. increase female arousal (essentially acting as a triggering super stimulus) and B. attract females from a distance (cumulative amplitude). Males here are benefitting from drawing females to a set location and potentially contributing to their general arousal. The primary cost here is presumably that the more males congregate and the less any individual male’s behavior matters, the more breeding becomes related to chance/social dynamics. If one could go off alone and still reliably attract females and inspire breeding interest in them then presumably that would be a better strategy. Here the physiological cost of the signaling doesn’t seem very pertinent, because presumably by going to a set location and drawing females there is a big energy saver over hunting around for females in a complex environment.

This literature raises a few simple questions for the authors: 1. Are African penguins lek breeders? 2. What is the prevalence of acceleration in other lekking species’ calls?

This takes me to another primary concern (or at least need for clarification) - the authors note that their model converged on acceleration rates very close to that of the African penguin. That is interesting! But the African penguin is just one of many species with accelerating call structure (including at least 3 other penguin species!). I know the authors have worked on African penguin calls recently, so those data are already processed and available. But even without extended formal analysis, presumably there are some characterizations of acceleration rates in the literature on other species’ vocalizations. Does their model match those as well? Is the acceleration rate they modeled some kind of biological universal? If not, it seems a little troubling that so much weight is put in the current paper on the African penguin. The authors even state in the discussion “the agreement between real-world data and model results highlights the potential of African penguins as a model species for investigating accelerando.” But the logic here is totally backwards. The model is as good as its ability to predict behavior in a range of species. The African penguin is not validated as a model species because its behavior matches a simple evolutionary theoretical model. Rather, this match leans in the direction of validating the model, but only if the model also reasonably matches the behavior of other species under similar pressures.

So, how were the assumptions of the current model set? How were benefit and cost determined? Did they take into account features of African penguin biology/behavioral and social ecology? If not, are they general assumptions that should hold for a wide range of species? I think these points need to be addressed up front in an accessible manner for non-modelers.

A few other points of moderate importance:

I recognize that this is likely too complex for the current model, but the proximal mechanism of call acceleration is quite interesting to consider. In general, arousal states predict call rates across a range of species (again, just the tip of the iceberg: Borjon, J. I., Takahashi, D. Y., Cervantes, D. C., & Ghazanfar, A. A. (2016). Arousal dynamics drive vocal production in marmoset monkeys. Journal of neurophysiology, 116(2), 753-764. Hausberger, M., Giacalone, A., Harmand, M., Craig, A. J., & Henry, L. (2020). Calling rhythm as a predictor of the outcome of vocal interactions: flight departure in pale-winged starling pairs. The Science of Nature, 107, 1-7.). This suggests that evolution may be acting on arousal during calling or the intersection between them. The ecstatic call is a breeding display, which makes this even more plausible. Further, the ecstatic call often functions as a mutual at-nest display between male and female (Eggleton P, Siegfried WR (1977) Displays of the Jackass Penguin. Ostrich 50 (3) 139–167), which seems likely to bear heavily on its function and timing aspects. All that to suggest that it’s possible that it’s rapid arousal spikes that evolution is working on here, and the call characteristics are derivative of/epiphenomenal to that.

Perhaps I’m missing the argument here, but is it plausible to suggest that call onsets are wholly phase-blind/irrelevant? That is, is it plausible to suggest a random onset time? Because calls frequently trigger other nearby birds to call it would seem that it is in fact NOT plausible. The timing could be variable, but if one call triggers another there is, by definition, a relationship between onset times. I don’t know how that would impact the modeling, and that is of course separate from dynamic phase adjustment, which is likely much rarer in animal vocal interaction. But a little clarification here would be helpful. This would seem to hugely affect the likelihood of overlap in the model depending on assumptions/lack thereof.

Obviously MORE vocal behavior has higher metabolic cost than LESS vocal behavior, generally speaking. I did wonder about the specific metabolic-cost assumptions here. If arousal drives calls, and arousal is necessary for breeding/nest defense, perhaps the vocal cost is minimal on top of the general physiological necessity of increased arousal in these contexts. I was also wondering about the physiological dynamics of calling in this species – sometimes faster rates are, paradoxically, less physiologically costly (e.g., running downhill vs walking, or, something else the authors have examined, the dynamics of horse gates).

Many of my primary concerns boil down to this: Why THIS model? Part of the issue is just, what are the modelers’ theoretical priors? Because it seems plausible that one could make a great many different models that showed acceleration was beneficial, and by tweaking model parameters get the actual rate to fall somewhere in the range of any particular species. What is the core theoretical commitment in this model that could be tested by comparison to data from different real world species?

A couple of minor points:

Paragraph starting on line 100 doesn’t really make sense. It starts w/ a fair point - the benefit of overlap might seem counterintuitive. But then it does nothing to suggest potential benefit of overlap. The rhetorical question about why animals should care to detect synchrony if its an epiphenomenon doesn’t clearly integrate into the rest of the paragraph. Should be looked at closely/clarified.

Line 392 - what is meant by “higher than average overlap”? Compared to other species? Based on what assumptions/characteristics of banded penguins?

Reviewer #5: To explore the evolutionary connection between rhythmic regularity and acoustic overlap, the authors developed a game-theoretical model to investigate whether rhythmic accelerando could evolve under selective pressure for acoustic overlap. This approach is based on two plausible key assumptions:

1. Acoustic overlap can provide evolutionary advantages in specific contexts or ecological settings.

2. Higher rates of acceleration and faster tempos impose greater costs on individuals.

The payoff matrix shows that acceleration strategies outperform diagonal strategies. Additionally, the authors conducted computer simulations to determine the mutation-selection equilibrium distribution under the assumption of a low mutation rate. These simulations further validate the evolutionary advantage of rhythmic accelerando and align closely with empirical data from African penguins. In the discussion section, the authors thoroughly elaborate on the broader significance of their findings.

In the previous round of review, the authors provided detailed responses to reviewers' comments and implemented the suggested revisions. Therefore, I recommend that this manuscript be accepted for publication.

**Have the authors made all data and (if applicable) computational code underlying the findings in their manuscript fully available?**

Reviewer #1: Yes

Reviewer #4: Yes

Reviewer #5: None

PLOS authors have the option to publish the peer review history of their article (what does this mean? ). If published, this will include your full peer review and any attached files.

**Do you want your identity to be public for this peer review?** For information about this choice, including consent withdrawal, please see our Privacy Policy .

Reviewer #1: **Yes: ** Andreas Baumann

Reviewer #4: No

Reviewer #5: No

---

## [Editor Report · Acceptance letter]

PCOMPBIOL-D-24-01329R1

An evolutionary model of rhythmic accelerando in animal vocal signalling

Dear Dr Jadoul,

I am pleased to inform you that your manuscript has been formally accepted for publication in PLOS Computational Biology. Your manuscript is now with our production department and you will be notified of the publication date in due course.

With kind regards,

Lilla Horvath
